# Deep Learning for Video Application in Cooperative Vehicle-Infrastructure System: A Comprehensive Survey

## Beipo Su, Yongfeng Ju and Liang Dai *

School of Electronics and Control Engineering, Chang'an University, Xi'an 710054, China; 2019032001@chd.edu.cn (B.S.); yfju@chd.edu.cn (Y.J.)
* Correspondence: ldai@chd.edu.cn; Tel.: +86-136-0929-3915

**Abstract:** Video application is a research hotspot in cooperative vehicle-infrastructure systems (CVIS) which is greatly related to traffic safety and the quality of user experience. Dealing with large datasets of feedback from complex environments is a challenge when using traditional video application approaches. However, the in-depth structure of deep learning has the ability to deal with high-dimensional data sets, which shows better performance in video application problems. Therefore, the research value and significance of video applications over CVIS can be better reflected through deep learning. Firstly, the research status of traditional video application methods and deep learning methods over CVIS were introduced; the existing video application methods based on deep learning were classified according to generative and discriminative deep architecture. Then, we summarized the main methods of deep learning and deep reinforcement learning algorithms for video applications over CVIS, and made a comparative study of their performances. Finally, the challenges and development trends of deep learning in the field were explored and discussed.

**Keywords:** video application; CVIS; deep learning; generative and discriminative deep architecture; deep reinforcement learning





## 1. Introduction

In recent years, with the rise of the social vehicle ownership rate, the needs of traffic safety and quality of user experience (QOE) are increasing. Intelligent transportation systems (ITS) are expected to relieve traffic pressure and prevent traffic jams through video target detection, video-assisted driving and wireless resource management. The proposal of CVIS can greatly alleviate the pressure of network overheads in ITS. Vehicles equipped with wireless communication units and sensing units are used as mobile nodes, and their communication modes include vehicle-to-vehicle (V2V), vehicle-to-infrastructure (V2I) and vehicle-to-roadside (V2R) [1] modes. Video applications over CVIS is one of the important parts of ITS. These were divided into video transmission, video content distribution and video target detection, which can effectively grasp road conditions, improve driving safety and provide users with popular video streams to improve the QOE. However, this attention-worthy problem ensures the accuracy of target detection, the low delay of video transmission and the high quality of video streams in a CVIS with the characteristics of a dynamic network topology, small transmission radius and time-varying channel.

As one of the current popular research fields, deep learning was first proposed by Hinton in 2006 [2]. It shows excellent performance in dealing with large amounts of data and high computational complexity. Through the nonlinear transformation of a multi-layer network structure, shallow features are combined to extract deep abstract features and realize the distributed representation of data. Because of its shallow structure, the traditional algorithm has difficulty obtaining the essential features from high-dimensional datasets in complex traffic environments. Researchers have applied deep learning to solve this problem, because of the advantage of the deep network structure, which can extract

the essential features of high-dimensional datasets. This paper summarized the research of deep learning in the field of video applications over CVIS in recent years, compared and analyzed the existing research methods, and discussed the current challenges and prospects for the future of video applications over CVIS based on deep learning.

This paper summarized the main research methods in the field of video applications over CVIS, and classified them based on the characteristics of their research methods, as shown in Figure 1.

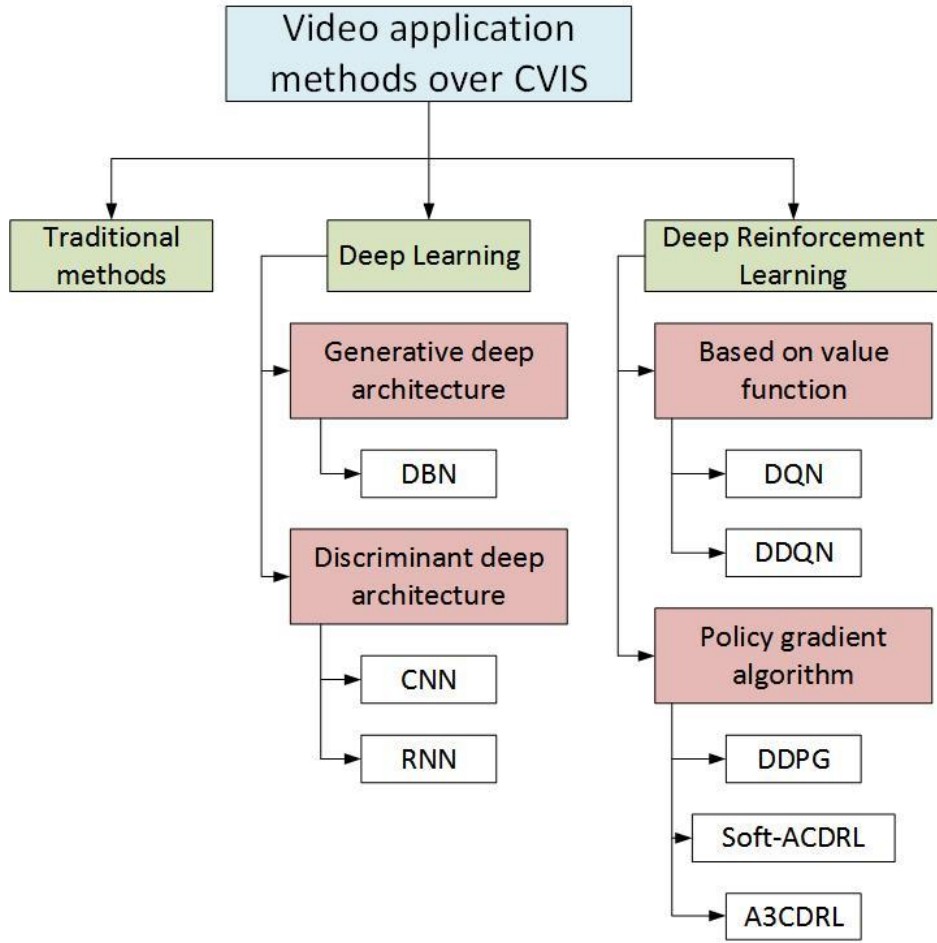

**Figure 1.** Video application methods over CVIS.

The rest of this paper is organized as follows. Section 2 analyzes the advantages and disadvantages of traditional methods for solving video application problems in vehicle–road coordination systems. Section 3 focuses on addressing the current challenges faced by video applications through deep learning. Section 4 expounds the solution of video application problems in complex CVIS through deep reinforcement learning. Section 5 presents the datasets and model performance evaluation metrics. Section 6 discusses potential future trends in the field.

## 2. Traditional Video Application Methods over CVIS

Video applications play a very important role in vehicle driving assistance, vehicle safety applications and entertainment applications. Due to the intelligence of the roadside infrastructure in CVIS, it is equipped with information storage units, computing units, etc., which can interact with vehicles in real time, and greatly promotes the development of video applications. In recent years, many researchers have conducted in-depth research on the use of video applications over CVIS and have proposed a considerable number of video transmission methods and video target detection methods.

Modern video coding standards can be divided into the following categories: H.264/AVC, H.264/SVC, H.265/HEVC, AVS2 and AV1 [3–5]. At present, the encoder optimization algorithm based on the H.26X standard is widely studied, and it can reduce the video freeze time and computational complexity under the condition of ensuring video quality in complex CVIS environments. The extensible modulation (s-mod) scheme proposed by Lin et al. [6] ensures the performance while significantly reducing the complexity of the encoder optimization algorithm. Belyaev et al. [7] reduced the computational complexity through unequal packet loss protection and rate control algorithms for scalable video coding. Zhang et al. [8] proposed a video layering scheme based on the H.265/SVC standard to improve the average offload traffic.

Forward error correction (FEC) is widely used in video coding, and can increase the credibility of data communication. Generally speaking, it causes a waste of channel resources when the receiver has no right to request retransmission in unidirectional communication after an error occurs in the transmission data. FEC is an error control mode which encodes according to an algorithm set in advance, before the data transmission, and adds redundant code information with its own characteristics. When an error occurs in the data transmission, the receiver decodes the received data stream according to the corresponding algorithm and determines the resulting error code to correct it. Ali et al. [9] used RaptorQ code as an FEC scheme to reduce data packet loss and improve transmission performance. Sofiane et al. [10] proposed an enhanced adaptive sub-packet forward error correction (EASP-FEC) video stream transmission scheme which aims to improve the quality of video transmissions over CVIS. To minimize video distortion, Zhu et al. [11] proposed an adaptive truncation hybrid automatic repeat request (ATHARQ) algorithm to find hierarchical video packet scheduling for interlayer forward error correction (IL-FEC) coding.

Resources can be divided into network resources and cache resources based on different standards. According to the different forms of resources, the means of resource allocation and scheduling will be different.

The cluster methods allocate video resources to vehicles with the same demand characteristics. Aiming to minimize video interruption probability, Jetendra et al. [12] showed a video resource decision scheme based on a cluster coverage and fast handoff mobile IP system (COMIP). Yaacoub et al. [13] divided mobile vehicles into cooperation clusters to improve the quality of service (QOS) and quality of experience (QOE), and proposed a V2V cooperative communication method.

The main joint DSRC/LTE communication method is that vehicles are allocated a corresponding communication channel according to different needs. Ben et al. [14] proposed a hybrid communication method, based on DSRC/LTE, which improves the overall reliability of communication. Xiaoman et al. [15] proposed a service-aware wireless access technology (RAT) selection algorithm to improve performance.

The multipath transmission method means allocating video sub-resources to different links for transmission. Xie et al. [16] proposed a multipath solution based on a disjoint algorithm to increase the transmission rate and reduce the delay. Aliyu et al. [17] proposed an interference-aware multipath video streaming (I-MVS) framework which aims to reduce packet error rate.

A method to reduce transmission cost, backhaul burden and transmission delay is named video cache transmission. It mainly caches the video to the roadside unit (RSU), and then the RSU transmits the video stream to the user. Guo et al. [18] proposed a dual-time-scale dynamic cache scheme to reduce the backhaul burden and improve video quality. Liu et al. [19] proposed a cache-based cooperative video transmission scheme in cellular networks which improves the QOE. Zhikai et al. [20] proposed an active video content caching (RCC) scheme to minimize transmission delay. Sun et al. [21] designed cache placement and short-term transfer strategies for long-term-aware transmission to improve the QOE.

Video target detection is mentioned to solve such problems as predicting a traffic situation, tracking a trajectory and identifying vehicle types, so as to improve traffic convenience

and driving safety. Ravi et al. [22] collected target parameters through video graphics detection, and predicted vehicle delay, which is defined as the extra time the vehicle spends at an intersection, by using support vector machine (SVM) and artificial neural network (ANN) algorithms. Comparing the two algorithms, it was concluded that ANN is the best-fitting model for estimating the delay of signalized intersections. Olayode et al. [23] obtained datasets from traffic data equipment, such as cameras, at seven road intersections of the most congested roads in South Africa, and developed an ANN model that divides these data sets into thirteen inputs and one output. The results showed that the accuracy of the ANN prediction method is effective.

The traditional video transmission method can show good performance in a simple CVIS environment. However, the algorithm of the traditional method converges slowly, and the model optimization is more difficult, due to more interference and the large data scale in complex environments.

## 3. Video Application Based on Deep Learning Algorithm

CVIS has the characteristics of a dynamic topology and a time-varying channel state. The traditional video application methods can not accurately analyze and extract the deep features of traffic data. The computational complexity of the traditional method is significantly improved, resulting in dimensional disaster [24] when the dimension of the dataset is high. By taking into account relatively few variables, the traditional methods can not reflect the real traffic characteristics of the CVIS environment. In addition, the traditional method's model is relatively simple, and it is easy to find the local optimal solution. However, deep learning can deal with these problems very well.

Deep learning is an important branch of machine learning, which originates from the study of artificial neural networks and is a further deepening of the basis of traditional neural networks. Shallow networks, such as most classification and regression algorithms, are limited to showing good performance in dealing with complex high-dimensional problems when the sample size is not large enough and the computing power is weak. On the other hand, deep learning mainly completes the complex function approximation through a deep nonlinear network structure, and can extract an abstract distributed representation of its essential features from a small number of samples [25,26]. Hinton et al. [2] proposed a fast learning algorithm based on a deep belief network (DBN) which is mainly divided into two steps: firstly, unsupervised learning is used to pre-train the constrained Boltzmann machine (RBM) network at each layer, and then supervised learning is used to fine-tune the whole DBN. This model reduces the difficulty of deep structure optimization, so the development and application of deep learning are paid increased attention by researchers. Because of the differences in the structure and training methods, deep learning can be divided into the following three categories:

1.  Generative deep architecture: this architecture can be described as a model for generating data which belongs to a probability model. Through the joint probability distribution of the observed data and the corresponding categories, the feature set of the generated data contains the high-order correlation features of the input dataset, which is more like an unsupervised learning method;
2.  Discriminant deep architecture: this architecture classifies patterns through its own discriminant ability, which estimates a posteriori probability through the conditional probability distribution of the observed data, similar to a kind of supervised learning;
3.  Mixed deep architecture: this architecture combines the advantages of generative and discriminative deep architecture, and has excellent expression ability and discriminant ability.

Based on the above three architectures, the advantages of algorithms in the field of video application over CVIS were summarized, respectively. At present, most of the research methods in the field are based on the generative and discriminant models, and the research on the hybrid model is still lacking. In addition, deep reinforcement learning

also promotes the further development of deep learning, so this paper also summarized the research in this field.

### 3.1. Generative Deep Architecture

DBN is a probability generation model which is obtained by a multi-RBM stack and layer-by-layer greedy training. RBM is a shallow neural network model based on energy. Its structure, shown in Figure 2, consists of a visible layer and a hidden layer. The nodes between layers belong to two-way connections, while the nodes within layers do not produce connections. The visible layer, also known as the input layer, is used to describe data features. The hidden layer will not send and receive signals through the outside; its function is to extract the abstract dependency relationship between data features so that it can be better linearized.

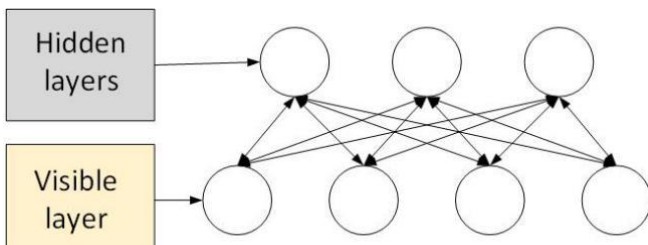

**Figure 2.** RBM structure schematic diagram.

Because of the structure of multi-layer neural networks, deep neural networks lead to some problems, such as a low learning efficiency, a high dataset sample size and a difficulty with parameter selection. However, DBN can deal with the above problems well by its novel training methods, so deep learning has been widely studied and applied.

The existing routing and caching strategies in vehicular networks cannot effectively cope with time-varying network conditions; thus, Zhang et al. [27] proposed an intelligent routing algorithm based on DBN (IRA) which aims to provide high-quality video streaming services in vehicular networks. Its model structure is shown in Figure 3. In order to discover the deep abstract relationship between the order of requesting video content and the time spent requesting video content, the DBN-based learning model [28] optimizes itself through back propagation regret after layer-by-layer pre-training (regret is defined as the difference between the predicted results and the actual results). Simulation results show that IRA achieves better network performance than other algorithms in terms of lookup data delay and cache hit rate.

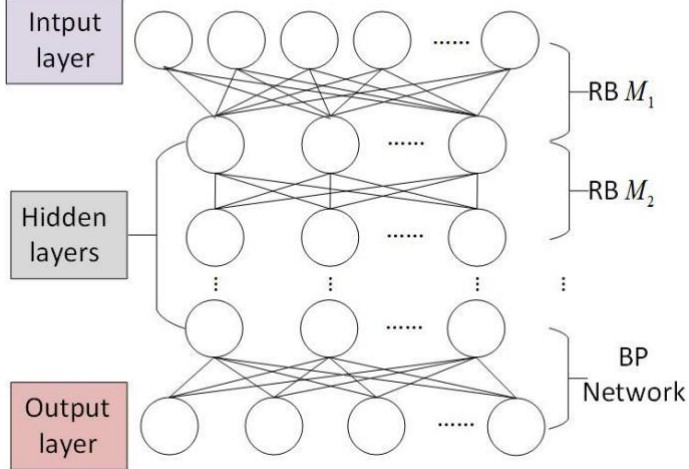

**Figure 3.** Structure of DBN.

### 3.2. Discriminant Deep Architecture

The models proposed by researchers based on the discriminant deep architecture include convolutional neural networks (CNNs) and cyclic neural networks (RNNs) in the field of video applications over CVIS.

### 3.2.1. CNN

The model output of a CNN [29,30] has no feedback connection with itself, so it is a kind of feed-forward deep network. It consists of an input layer, a convolution layer, a pooling layer, a full connection layer and an output layer.

The input layer is used mainly to preprocess the original data, to reduce the impact of differences. As the most important layer of a CNN, the convolutional layer is composed of multiple feature graphs, each of which is composed of multiple neural nodes. The convolution kernel is a weight matrix: the convolution effect of convolution kernels of different sizes will be different when the extraction target is the same. The convolutional layer extracts the specific features of all nodes on the feature graph through the sliding convolution kernel, and the sliding step is expressed as the length of a sliding of the convolution kernel. Each neural node in the convolution layer is locally connected with the feature graph of the corresponding input layer through a set of weights [26]; then, its local weighting sum is mapped by a nonlinear function to obtain the data output of the convolution layer. Importantly, its weights are shared when extracting feature data on the same feature graph. The pooling layer is the output layer of the convolution layer, which is also composed of multiple feature graphs, and each feature graph corresponds to the convolution layer one by one. Average pooling and maximum pooling are common methods in this condition which can reduce the resolution of the feature surface while ensuring that the feature data will not be affected by spatial changes [31], reduce the dimension of the feature data by removing redundant information and prevent over-fitting to a certain extent. The purpose of pooling is to reduce the computational complexity. The connection layer's purpose is to integrate local information with category distinctions, after alternating multiple convolutional and pooling layers [32].

It becomes very difficult to deal with the problem of image distortion when encountering bad weather such as heavy fog and rainy nights. Image quality assessment [33] is mainly used to predict the perceptual quality of digital images, and CNN models are found to be robust to distortion [34]. Varga, Domonkos [35] proposed a depth-based no-reference image-quality assessment architecture that incorporates multiple CNN models, which effectively evaluates image quality by considering multiple image quality scores from different CNN models. This architecture was confirmed in experimental tests.

Chen et al. [36] described a new method, based on deep learning, which removes multiple error sources in sensor signals such as cameras at the same time in the laboratory environment. By correctly identifying the classified signal, the CNN algorithm is developed, which essentially eliminates the sensor error source. In order to test the efficiency of the algorithm, the results are compared with the traditional six-bit static test, the rate test and other methods. An accuracy of 80% is achieved in correctly identifying accelerometer and gyroscope signals.

Kumar et al. [37] introduced a case study of a CNN-based solution using camera video data for real-time open-air off-street intelligent parking management. Experiments were carried out on real-time 24 h data from the input camera video source installed in the parking lot of IIT Hyderabad (IITH). The experimental results show that this scheme can improve parking performance.

Jeon et al. [38] proposed a scene representation method, that is, a multi-channel occupation grid graph (OGM) to describe the whole traffic scene. The deep learning architecture of the OGM is used to predict the future traffic scene. By using this 2D traffic scene representation, the future prediction can be modeled as a video processing problem in which the future time series image series needs to be predicted. In order to predict the future traffic scene according to the past traffic scene, a deep learning architecture using

a CNN and long-term short-term memory network is proposed. In the case of highly conflicting traffic, the future prediction accuracy can be as high as 90%, and the prediction range is 3 s by using the proposed deep learning framework.

Akilan et al. [39] proposed a multi-view perceptual field codec convolutional neural network (MvRF-CNN) model, which contributes mainly to the use of multiple views of convolution kernels with residual feature fusions in the early, middle and later stages of the codec (EnDEC) architecture. The model is still effective in complex traffic environments (dynamic background, camera jitter, night video, bad weather, etc.).

Shobha et al. [40] proposed a deep learning adaptive active network subdivision model for vehicle division. The proposed solution includes three stages: subtraction based on an adaptive background model, active network subnet using a CNN and optimization using an extended topology active network (ETAN) to extract data from the CNN results. Adaptive background modeling is based on the adaptive gain function, which is composed of pixels of frames in the video. The gain function can compensate for the shadow and lighting problems that affect the vehicle division. The deep learning-assisted topology active network deformable model can provide higher partition accuracy in the presence of occlusion, cluttered background and traffic density changes.

Ma et al. [41] proposed an intelligent collaborative visual perception system for networked vehicles. The driving video is collected from the vehicle and transmitted to the cloud for visual perception using a CNN. In order to receive and process multiple vehicle videos synchronously, several data pipeline CNN frameworks are developed; their frame structures are shown in Figure 4. Considering the bandwidth consumption and transmission delay, the resolution adaptive strategy and frame rate control strategy are designed, and IPv6 routing is used to transmit between the vehicle and the cloud. The system helps drivers to make better and safer decisions by enhancing their perception.

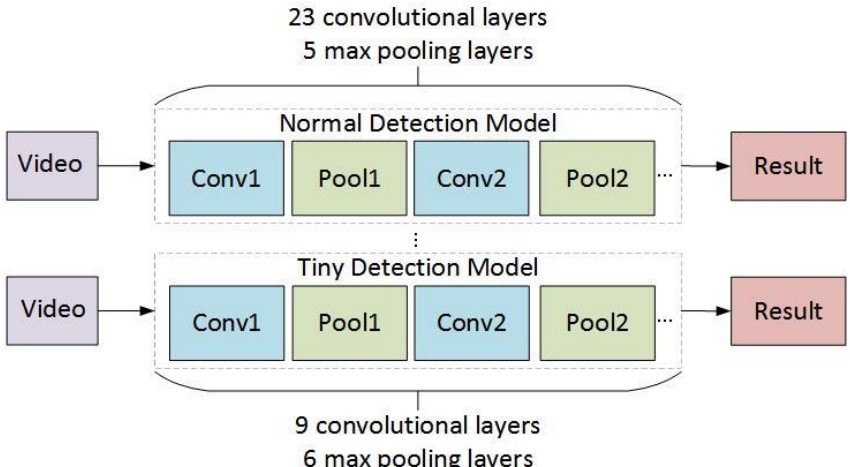

**Figure 4.** Multi-data pipeline CNN framework.

Guo et al. [42] combined video compression sensing (CVS) with a CNN, and proposed a correlation analysis model in the measurement domain called CVS-CNN. They used CNN instead of the pseudo-inverse transformation of the measurement matrix, and established the correlation between the measured values of the blocks to be estimated and the measured values of adjacent non-overlapping blocks, which can analyze the time correlation of video frames in the measurement domain. The processing speed, accuracy and robustness are obviously better than similar video frame correlation analysis methods.

Jeong et al. [43] proposed an intrusion detection method based on feature generation and CNN for the first time to detect attacks on audio and video transmission protocol (AVTP) streams in a network based on automobile Ethernet. In order to evaluate the intrusion detection system, a physical test platform based on BroadR-Reach and captured real AVTP packets was built.

Madhumitha et al. [44] proposed a new heuristic unimodal method based on a vision system to estimate the collision priority of vehicles on the road. Priority is estimated from the point of view that it may be equipped with a vision-based driver assistance system or the self-driving vehicle itself. Consider crowdsourced videos from YouTube about vehicle collisions captured by vehicle onboard cameras: in order to detect the moving vehicle in the video, the pre-training object detection model and tracking algorithm based on CNN are used.

Wang et al. [45] proposed a robust hierarchical deep learning for promoting congestion detection. In this method, a deep network is designed for hierarchical semantic feature extraction. Different from the traditional deep regression network, which usually uses the mean square error directly as the loss function, robust metric learning is used to train the network effectively. On this basis, multiple networks are combined together to further improve the generalization ability. Extensive experiments have been carried out and it is proved that the proposed model is effective.

A region-based CNN algorithm (RCNN) was proposed by Girshick et al. in 2014 [46]. The author applies a CNN to the field of target detection for the first time. The algorithm is mainly composed of four parts: candidate region generation, feature extraction, category judgment and location refinement. Because of its excellent performance, RCNN is widely used in the field of video applications over CVIS.

Priyadharshini et al. [47] proposed an RCNN algorithm to deal with the aggregation of vehicle data from real-time video streams which improves the detection accuracy by 3.5% compared with a CNN. Kamran et al. [48] prepared datasets containing real data (taken from military performance videos) and toy data (downloaded from YouTube videos). The datasets were divided into three main types: military vehicles, non-military vehicles and other non-vehicle objects. In order to analyze the adequacy of the prepared datasets, we use algorithms such as RCNN target detection to distinguish between military and non-military vehicles. The experimental results show that using the customized/prepared datasets to train the deep architecture can identify seven kinds of military vehicles and four kinds of non-military vehicles.

You Only Look Once (YOLO) was proposed by Joseph et al. in 2015 [49]. YOLO is a target detection algorithm based on CNN, which means that target features can be detected only once. Because of this feature, YOLO is favored by researchers in the field of real-time video target detection. Seal et al. [50] devoted themselves to developing benchmark applications for real-time traffic incident identification and related traffic management using real-time congestion-aware navigation of intelligent vehicles with video feeds, proposing an algorithm framework based on YOLO for identifying and classifying traffic events, such as traffic accidents and congestion and building a fog layer between edge nodes and clouds to make distributed computing (servers) and provide storage closer to edge nodes. Experiments show that the total latency or response time of the YOLO-based fog–cloud platform is lower than that of the pure cloud platform.

Pham et al. [51] proposed a target detection framework based on YOLO to detect and track a large number of highly mobile vehicles, which is also considered as the region of interest (ROI) in vehicle optical camera communication (OCC) systems. The author tested the method on a rainy night on a Korean highway to analyze the effectiveness of this method in the vehicle OCC system. Humberto et al. [52] proposed a real-time statistical algorithm framework for YOLO vehicle classification based on edge AI. Experiments show that the algorithm is helpful to provide timely traffic information.

Song et al. [53] aimed to use real-time traffic data collected by traffic surveillance video and image recognition to explore the relationship between the spatio-temporal pattern of vehicle types and numbers in different urban functional areas and the emission of traffic-related air pollutants. Video-based YOLO detection technology is used to analyze the data, and air pollution is quantified by pollutant emission coefficient.

Sreekumar et al. [54] proposed a deep learning model based on YOLOv2, and applied the model to traffic detection scenarios. In order to reduce the network burden and eliminate

the deployment of the backbone of the network at the intersection, it is recommended that the traffic video data be processed at the edge of the network without transmitting big data back to the cloud. In order to improve the frame rate of edge processing, this paper further proposed a depth object tracking algorithm based on an adaptive multimodal model which is robust to object occlusion and changing lighting conditions.

Huang et al. [55] aimed to develop an efficient system that can meet the needs of video analysis based on dynamic content and expand to large-scale traffic camera video data. The proposed system used a YOLO-based deep learning method to identify objects in video data. This information was then processed and analyzed by the analysis layer implemented using Spark and Hive. Compared with the traditional method, the accuracy of this system can reach more than 80%.

### 3.2.2. RNN

There is a correlation between the data when the data has sequence characteristics and the accurate feature representation cannot be obtained by analyzing and extracting features from a single data. In order to solve this problem, the RNN was proposed by Pineda et al. [56]. When an RNN is faced with time series data, according to its special network structure, the information of the previous time can be mapped non-linearly to the next time, and the dependency relationship in the data is maintained, which gives RNN the ability to remember and process the previous data information so that it can extract features from the sequence data. For RNN, its weight parameters are shared during a training process.

After follow-up practice, it is found that it is difficult for RNN to deal with long-term and long-distance dependence problems, which often lead to RNN gradient disappearance or gradient explosion [57]. The long-term short-term memory (LSTM) model proposed by Hochreiter et al. [58] can deal with the above problems well. The structure of LSTM is relatively complex, and its memory unit includes three kinds of gating units: the forgetting gate, the input gate and the output gate. The forgetting gate can filter the historical data and discard the information that feels useless in the historical information, the input gate can control the storage of part of the current data into the memory unit and part of the stored information in the output control memory unit is output at the current time. Because of the above features, LSTM can better store useful information and plays an important role in solving the problems of long-term and long-distance dependence. As a further study of LSTM, gated cyclic unit neural networks (GRUs) couple the input gate and the forgetting gate into a gated unit, which means the reduction of parameters and the reduction of computational complexity. Therefore, compared with LSTM, GRU has better training efficiency.

Adita et al. [59] designed a sequence-to-sequence deep learning model called DeepChannel that is based on a codec. It can predict the change of wireless signal strength in the future according to the past signal strength data. This article considers two different versions of DeepChannel; the first and second versions use LSTM and GRU as their basic unit structures, respectively. Different from the previous work of designing models for specific network settings, DeepChannel has strong adaptability and can predict future channel conditions for different networks, sampling rates, mobility modes and communication standards.

Yang et al. [60] proposed a unified driver behavior modeling system for multi-scale behavior recognition. The driver behavior recognition system aims to identify the physical and mental state of the driver simultaneously based on the deep encoder–decoder framework. The model learns to recognize three different time scales of driver behavior: mirror examination and facial expression state, as well as two psychological behaviors, including intention and emotion. The encoder module is based on CNN and is used to capture spatial information from the input video stream. Then, several decoders for different driver state estimations are proposed using RNN based on full connection (FC) and LSTM. The model framework is shown in Figure 5.

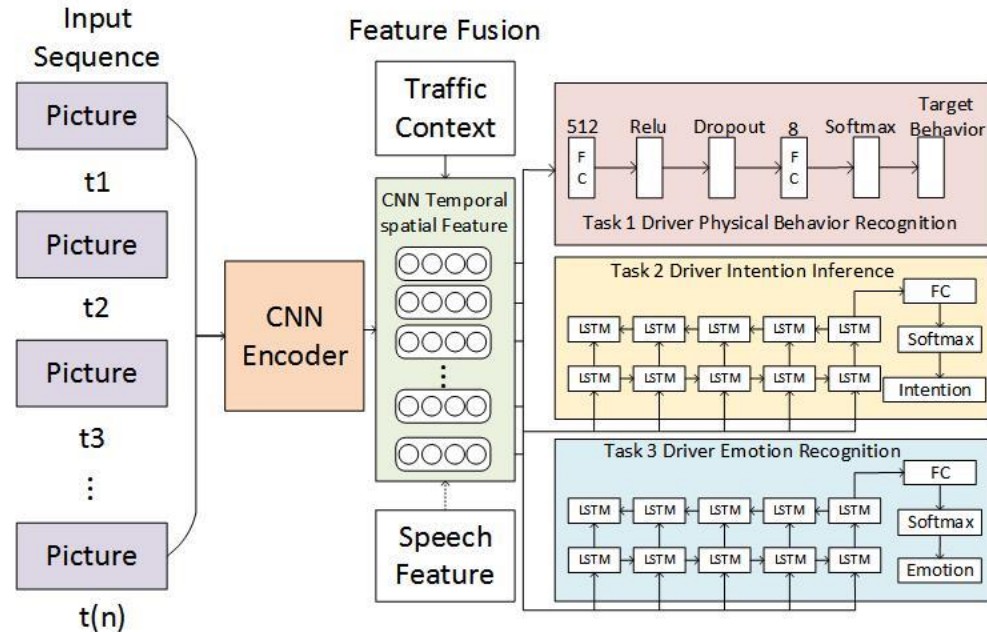

**Figure 5.** Multi-task learning model framework.

## 4. Video Application Based on Deep Reinforcement Learning Algorithm

Deep reinforcement learning is a combination of deep learning and reinforcement learning. With deep learning [26], through a gradual, layer-by-layer process, low-level data input is transformed into high-level abstract feature representations through nonlinear transformations, and deep learning pays more attention to the perception and expression of environmental feedback datasets. Reinforcement learning [61] obtains the optimal strategy to maximize the cumulative reward through the interaction between the agent and the environment. It focuses more on the strategies to deal with the problems through autonomous feedback learning. Deep reinforcement learning converts high-dimensional datasets of environmental feedback into abstract state features with the help of deep learning. Then, the optimal strategy is obtained with the help of reinforcement learning autonomous learning, which is used to solve the strategy problem in a complex environment.

In the video application environment over CVIS, the complexity of the environment is influenced by traffic flow, the speed of the vehicle, wireless channel resources, signal strength and so on. The environmental state space is often high-dimensional, and traditional reinforcement learning algorithms have difficulty dealing with high-dimensional state space and continuous action space. With the performance of deep learning in high-dimensional data input processing, the above problems can be easily solved. Therefore, deep reinforcement learning has gradually become the most-covered direction of researchers in the field of video transmission under CVIS. According to the solution strategies based on different methods and numbers of agents, they are divided into the following three categories:

1. Deep reinforcement learning algorithm based on value function: the algorithm mainly evaluates the Q-value generated by all actions, selects the action according to the Q-value and obtains the optimal strategy indirectly through the value function;
2. Policy gradient algorithm: the algorithm parameterizes the strategy, uses the weight parameters of the depth neural network to represent the strategy, optimizes the strategy through the gradient method and constantly modifies the parameters and gradually obtains the optimal strategy. The policy gradient algorithm is an algorithm to solve the optimal policy directly;
3. Multi-agent deep reinforcement learning: multiple agents choose the corresponding actions according to the current environment, establish different reward functions according to the relationship between agents and solve the optimal strategy.

Through the above three ways, this paper divided and classified the video application algorithms over CVIS based on deep reinforcement learning, and its characteristics and advantages were summarized.

### 4.1. Deep Reinforcement Learning Algorithm Based on Value Function

After the Deep Q-Network (DQN) was first proposed by Mnih et al. [62] in 2013, deep reinforcement learning has attracted increased attention. With the deepening of video transmission research over CVIS, DQN is widely used to solve video business-related problems.

Because of the large dimension of state space, the Q-value of traditional reinforcement learning is difficult to predict. The deep neural network can extract eigenvalues through multiple network layers to reduce the dimension, so DQN approximately represents the Q-value function through the deep convolutional network (DNN). The evaluated Q-value function may be unstable and deviated. The target network and the experience playback mechanism proposed by Mnih et al. [63] solve this issue.

In order to improve the revenue of RSU operators, Ahmed et al. [64] proposed a scenario in which vehicles covered by an RSU cache off-the-shelf content (video streams, etc.) to the RSU, and the RSU then provides services for newly arrived vehicles, which is defined as an MDP. The author defined the environmental state as the vehicle location information, the request content and the available content information, defined the action as whether or not to distribute the content and defined the reward as the available income. The DQN algorithm is used to solve the MDP problem; the structure of the DQN algorithm is shown in Figure 6. First, observe the state space: DNN extracts the abstract representation through the state space, RSU selects the action to execute and the environment gives feedback (the vehicle delivery cost). The DQN model proposed by the author is superior to other methods in terms of total income, cost and service rate.

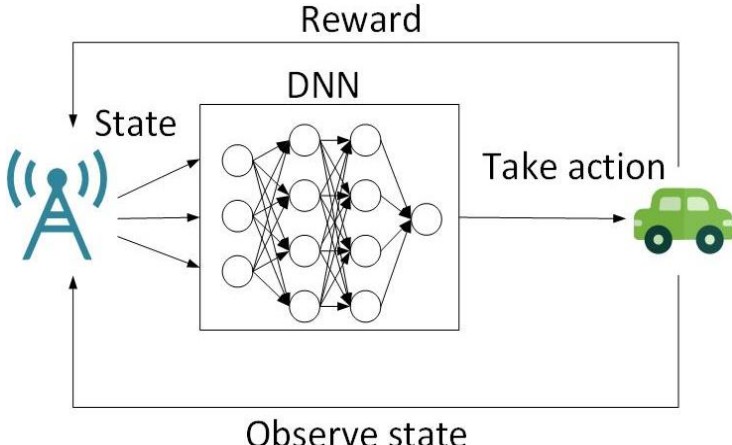

**Figure 6.** The structure of the DQN algorithm.

Atallah et al. [65] solve the security and QOS problems of green, balanced, linked and efficient vehicle networks. Using the DQN model, the completion request rate is increased by 10–25%, the average request delay is reduced by 10–15% and the total network life is increased by 5–65%.

Nassar et al. [66] developed a network slicing model based on the fog node (FN) cluster coordinated with the edge controller (EC) to solve the resource allocation problem. The author described the problem as an MDP, and the state space is defined as the resource block information, the action is defined as whether the edge controller sends the resource block to the corresponding FN cluster for processing and the reward is defined as the key performance indicator (KPI). Through DQN, self-adaptive learning is the best slicing strategy to maximize KPI, so as to achieve efficient resource allocation.

Pan et al. [67] proposed an asynchronous federated DQN-based and URLLC-aware computation offloading algorithm (ASTEROID) to maximize throughput under long-term

URLLC constraints. In this paper, a URLLC constraint model, based on extreme value theory, is established. Secondly, Lyapunov is used to optimize the decomposition of task unloading and computing resource allocation. Finally, an asynchronous federated DQN (AF-DQN) algorithm is proposed to solve the problem of vehicle-side task unloading. Its state space is defined as the backlog of information in actual and virtual queues, the action is defined as whether the task is unloaded and the reward is defined as the throughput. Simulation results show that ASTEROID achieves excellent throughput and URLLC performance.

Sun et al. [68] proposed a novel slicing framework and optimization solution based on dynamic reinforcement learning for efficient resource allocation in virtual networks with D2D-based V2V communication. The goal is to balance resource utilization and QOS satisfaction across multiple slices. The problem is described as an MDP whose state space is defined as the QOS satisfaction, resource utilization and the number of resource blocks occupied by each slice; the action is defined as the proportion of resources allocated to the slice and the reward is defined as the composition of resource utilization and QOS satisfaction. The optimal strategy is obtained by using the DQN model, and the performance results of the resource utilization, QOS satisfaction and throughput are given.

Tan et al. [69] studied joint communication, caching and computing design issues to achieve excellent operation and cost efficiency for in-vehicle networks. Because of the high complexity of CVIS, the author used the multi-time scale framework [70] to develop deep reinforcement learning to reduce the complexity. The multi-time scale framework consists of two models. The model for each cycle is called the large time scale DQN model, and the model for selecting the action decision for each time slot is called the small time scale DQN model. Numerical results are given to prove its performance gain.

To maximize the long-term sum ratio of all vehicles, Wang et al. [71] proposed a hybrid architecture which consists of centralized decision making and distributed resource sharing (C-Decision scheme). Each vehicle uses a deep neural network to compress its observed information, and then feeds it back to the central decision-making unit. The centralized decision-making unit uses DQN to allocate resources, and then sends the decision results to all vehicles. In order to promote distributed resource sharing, the author proposed a distributed decision and spectrum sharing architecture (D-Decision scheme) for each V2V link. Simulation results show that the proposed C-Decision and D-Decision schemes can achieve near-optimal performance and are robust to feedback interval variation, input noise and feedback noise.

Zhang et al. [72] studied the federated optimization of transmission mode selection and resource allocation for cellular V2X communication. The problem is expressed as an MDP whose state space is defined as the receiving interference power, the resource block neighbor information of the previous subframe, the channel gain, the current load and the remaining time to meet the delay threshold; the actions are defined as the resource block allocation, communication mode selection and transmission power level; the rewards are defined as the sum of the total capacity benefits, the unmet capacity penalties and the effects of reliability and delay requirements. A decentralized algorithm based on DQN is proposed to maximize the total capacity of vehicles to infrastructure users and to meet the delay and reliability requirements of V2V communication.

Hasselt et al. [73] proposed a deep double Q-network on the premise of an in-depth study of the double Q-learning algorithm. The traditional DQN may overestimate the Q-value, but DDQN reduced the overestimation by improving the training algorithm, so as to obtain a more accurate Q-value and optimize the performance of the model. Wang et al. [74] proposed a competitive network structure as a network model of DQN; its structure is shown in Figure 7. The competitive network structure divided the abstract features extracted by DNN into two places, in which the state value function represents the value of the environment itself, and its value does not change according to the action choice, while the action dominance function represents the additional reward value after the execution of the selected action. When the two are combined, the Q-value is generated

for each action. The duel DQN improves the prediction accuracy of the value function by improving the structure of the model.

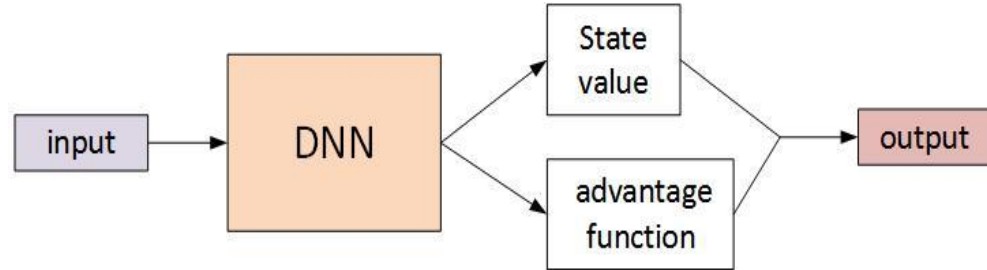

**Figure 7.** The DQN algorithm structure based on a competitive architecture.

In order to improve the performance of vehicle network, He et al. [75] proposed an integrated framework which can realize the dynamic allocation of network, cache and computing resources. The resource allocation strategy of the framework is described as a joint optimization problem. Because the state space is very complex, the traditional DQN algorithm has difficulty effectively approximating the Q-value function to obtain the optimal strategy. The author combined the deep double Q-network (DDQN) and the duel DQN algorithm to find the optimal strategy to solve the above problems. The effectiveness of the proposed scheme is proved by the simulation results of different system parameters.

Malektaji et al. [76] considered an edge content delivery network with vehicle nodes, and proposed using the DDQN model to solve the problem of content migration. The DQN model defined the state space as the priority information of the content and the delivery status of the content, the actions are defined as migrating, caching and deleting content and the reward is defined as the access delay. This model reduces content access latency by 70% compared with traditional policies.

Aiming to minimize the user cost, Nan et al. [77] studied the user-centered content delivery problem with service delay constraints in vehicle networking. The optimal content delivery strategy problem is modeled as an MDP which defines the state space as service time, vehicle location information, content cache status and signal strength; the action is defined as whether the vehicle obtains content delivery from the RSU or cloud and the cost is defined as the cost paid by the vehicle for content. The DDQN algorithm is proposed to solve the above problems, and can achieve the best performance while minimizing the cost.

*4.2. Policy Gradient Algorithm*

The complexity of the DQN seeks out the optimal policy by estimating that the Q-value indirectly increases exponentially when the dimension of continuous or discrete action space is large, so some researchers have focused on solving the optimal strategy directly. The stochastic policy gradient algorithm was proposed in 1999, which parameterizes the policy and updates the policy parameters by calculating the policy gradient. The deterministic policy gradient (DPG) algorithm, on the basis of a random policy gradient, was proposed in 2014. However, because all kinds of policy gradient algorithms need a large number of datasets to provide training—otherwise they will fall into the local optimal situation—the problem can be well solved by the combination of an AC algorithm and a policy gradient. The AC algorithm framework is shown in Figure 8. The AC algorithm is based on the idea of time difference (TD). The policy network is an actor who selects the action according to the environment state. The value function network is a critic who evaluates the action through the value function and generates the time difference error signal to update the improved policy network and the value function network.

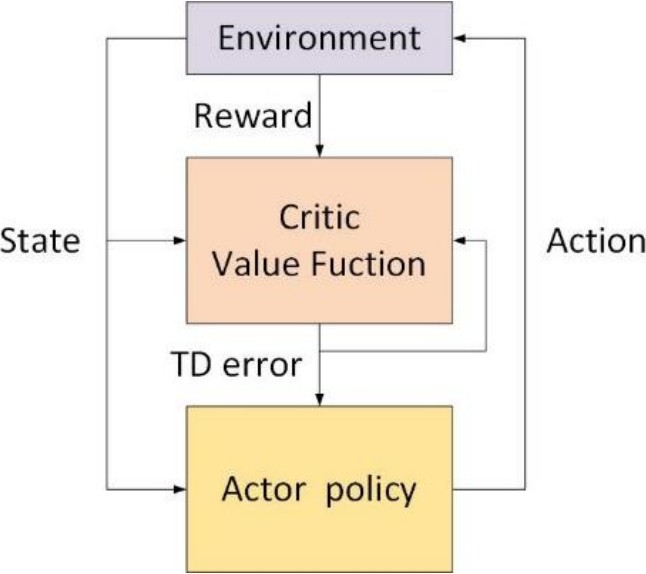

**Figure 8.** The AC algorithm framework.

Lillicrap et al. [78] combined a deep neural network with DPG for the first time in 2016, and proposed a deep deterministic policy gradient (DDPG) algorithm based on an AC framework, which makes up for the deficiency that DRL is unable to find the optimal strategy when it comes to continuous action spaces.

To minimize the content delivery latency, Dai et al. [79] designed an optimized vehicle edge caching and content delivery strategy using DRL. Edge caching and content delivery problems are described as MDPs; the state space is defined as an adjacent vehicle set, link transmission rate, cached content information and vehicle driving direction; the actions are defined as whether content is cached, the request content ratio and the bandwidth allocation; and the reward is defined as the content delivery delay. The DDPG algorithm is proposed to solve the above problems. Compared with the two benchmark solutions, the numerical results show the effectiveness of the proposed DDPG-based algorithm.

Fang et al. [80] aimed to maximize video quality while reducing delay and rate switching in traffic fog computing (VFC) systems by jointly optimizing vehicle scheduling, rate selection, computing and spectrum resource allocation. Considering the dynamic characteristics of the vehicle network and the available computing/spectrum resources, the problem is modeled as an MDP. Due to the action space of an MDP being a mixture of multi-dimensional continuous variables and discrete variables, this study uses a soft-ACDRL algorithm to solve the MDP problem, and its algorithm structure is shown in Figure 9. Lyu et al. [81] proposed a dual-scale DRL framework based on a soft-ACDRL algorithm to solve the problem of joint communication, computing and cache resource allocation, reducing costs and improving user satisfaction as a result.

The asynchronous advantage ACDRL algorithm (A3C) was proposed by Mnih [82]. Because of its multi-core CPU and multi-threading technology, the A3C algorithm makes multiple agents collect experience samples at the same time, and the correlation between samples is very low. It adopts the training mode of on-policy, and replaces the experience playback mechanism in the ACDRL algorithm by multi-line parallel and asynchronous updating parameters. The researchers have proved that the iterative speed of the algorithm is obviously better than that of the ACDRL algorithm. Jiang et al. [83] proposed a video analysis framework that integrates multi-access edge computing and blockchain technology into IoAV to maximize the throughput of blockchain systems and reduce the latency of MEC systems. The joint optimization problem is modeled as a Markov decision process (MDP) and is solved by the A3C algorithm based on DRL. Khan et al. [84] studied the correlation between vehicles and RSUs in millimeter wave (mmWave) communication networks. Their aim was to maximize the data transmission rate per vehicle user (VUE)

while ensuring that all VUE rates were not lower than the minimum rate and had a low signaling overhead. The association problem is expressed as an MDP, and a low-complexity algorithm based on A3CDRL, which is similar to the solution of the proposed optimization problem, is proposed.

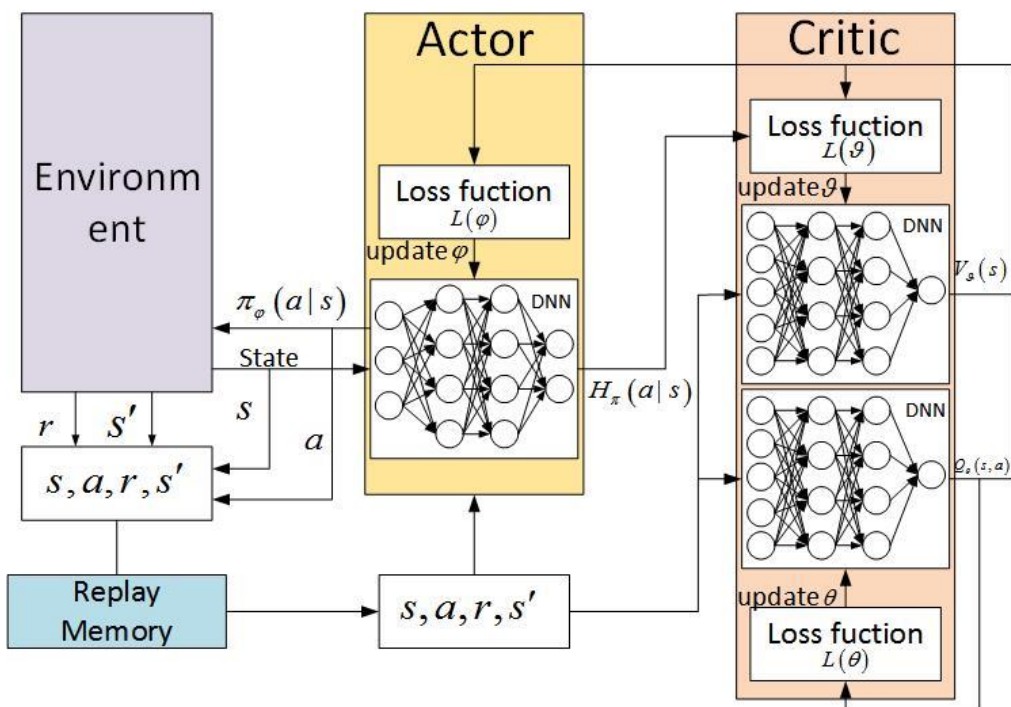

**Figure 9.** The structure of the soft-ACDRL algorithm.

As researching the DDPG algorithm is becoming an increasingly popular task, a large number of researchers have applied this algorithm to the field of video transmission over CVIS. This paper summarizes the research methods based on DDPG in Table 1.

*4.3. Multi-Agent Deep Reinforcement Learning*

The DDPG algorithm becomes helpless when faced with more than one agent. Tampuu et al. [85] proposed a DRL model of cooperation and competition among multi-agents, in which each agent is trained by independent DQN algorithm, and rewards are designed according to the goal of each agent. According to the different return functions designed, multi-agent deep reinforcement learning can be divided into three ways: complete cooperation, complete competition and mutual cooperation and competition. When the environment of video transmission over CVIS is so complex that a single agent cannot solve the problems, multi-agent DRL enters the attention of researchers.

Zhu et al. [86] proposed a computing offload scheme, based on multi-agent DRL, which minimizes the total task processing delay in the long run. In order to evaluate the performance of the proposed unloading scheme, a large number of simulations were carried out. Simulation results verify the effectiveness and superiority of the proposed scheme.

Amine et al. [87] studied the problem of service migration in MEC-enabled vehicle networks to minimize the total service delay and migration costs. In order to obtain an effective solution, it is modeled as a multi-agent MDP and solved by the deep Q-learning (DQL) algorithm. Simulation results show that the proposed scheme achieves near-optimal performance.

**Table 1.** Video application methods over CVIS based on DDPG.

| Author | State | Action | Reward | Research Content |
|---|---|---|---|---|
| Chen [88] | Unload target, computing resource information, available bandwidth and vehicle location information. | Whether to uninstall target and computing resources and available bandwidth allocation. | Task completion utility. | Intelligent allocation of edge computing resources and channel resources. |
| Dai [89] | Communication rate, energy consumption, driving direction, content delivery delay and storage capacity. | Match cache pair. | System utility. | Implement blockchain-enabling content cache security and privacy protection. |
| Kwon [90] | Preloaded video size, cache occupancy, average video quality and location information. | Power allocation and cache allocation. | Quality change, packet loss and video interruption. | Millimeter wave-based station power control and active buffer allocation method. |
| Lan [91] | Computing and caching capabilities. | Calculate the uninstall and service caching policy. | Task processing delay and energy consumption. | Optimization of joint computing offload and service cache. |
| Yun [92] | Location information, transmission queue status, receiving area status, number of blocks and average quality. | Whether to unload, unload quantity and block quality. | Quality enhancement, avoiding packet loss and video freeze. | Video streaming transmission scheme for mobile-aware vehicle network. |
| Zhang [93] | Video data transmission status, local view set status and vehicle association status with RSU. | A combination of the view set and memory allocation. | Download cost, data loss and video freeze cost. | Active caching of multi-view 3D video in 5G network. |

Kumar et al. [94] used deep learning technology to predict the movement pattern of vehicles. An architecture composed of a centralized decision-making unit and distributed channel allocation is proposed to maximize the spectrum efficiency of all related vehicles. In order to achieve the above goals, the authors used DQN and A2C technology and further integrated LSTM [58] into DQN and A2C technology. The performance of the proposed scheme is verified by simulation.

In order to solve the problem of the abnormal growth of computational complexity and the explosive growth of state space and action space in multitasking and multi-agent (MTMA) environments, Xu et al. [95] used a value decomposition mechanism (VDM) to decompose complex value functions into several small blocks, and then calculated each sub-function separately. The proposed paradigm can greatly accelerate the learning speed and help significantly in a smaller state and action space, but will not degrade the performance. Simulation results showed the effectiveness of the proposed mechanism.

## 5. Datasets and Model Performance Evaluation Metrics

Most video datasets are collected through realistic means. Generally speaking, the dataset is divided into a training set, a validation set and a test set. Specifically, in the training phase, the model parameters are trained and tuned using the training and validation sets, while the test set is used in the final testing phase. The performance evaluation metrics can more intuitively observe the advantages and disadvantages of the model. According to different optimization standards, it was divided into three types of performance evaluation indicators and listed the performance advantages of some distinct methods in video applications over CVIS based on deep learning. In this section, we summarize existing datasets and performance evaluation metrics for video applications.

### 5.1. Datasets

The full name of NGSIM is Next Generation Simulation, which is a data collection project initiated by the Federal Highway Administration that includes data on construction, vehicle trajectory prediction, driver intent recognition, automatic driving decision planning, etc. By

applying image processing algorithms to recorded traffic flow videos, Jeon et al. [38] construct a trajectory database of 18 variables related to vehicle state, and test the performance of its proposed deep learning architecture for predicting future traffic scenarios.

The COCO dataset contains 80 different object categories, and this dataset is used mainly for object detection, segmentation, etc. Seal et al. [50] used 2 CNN architectures to classify data: YOLOv3.11 (for 11 classes related to traffic congestion) for traffic object recognition; and YOLOv3.6 (for 6 classes related to traffic events).

Jeong et al. [43] tested its intrusion detection methods on its own carefully created automotive Ethernet intrusion dataset. The dataset was recorded in the format of a PCAP file so that it can be viewed using popular programming libraries and packet analyzers.

Shobha et al. [40] tested the performance of the proposed deep learning-assisted ETAN segmentation on the MIO-TCD dataset. The MIO-TCD dataset is a standard benchmark dataset for vehicle localization and identification. The dataset has about 500,000 images of vehicles taken from US roads at different times of the day. From this dataset, we selected 100 images to test the performance of the proposed work.

Ma et al. [41] used VOC2007, VOC2012 and videos captured in real driving scenarios, which contain 16,551 vehicles, buses, pedestrians, motorcycles and bicycles, to train the model.

### 5.2. Average Delay

The average delay can effectively reflect whether the video stream can be transmitted successfully in a limited time. The average delay includes mainly the average transmission delay, the average queuing delay, the average processing delay and the average propagation delay. When the video file is large, the article will consider primarily the average transmission delay and the average propagation delay. When the number of files is large, the article will consider the average queuing delay and the average propagation delay. The total latency or response time of the deep learning-based fog cloud computing framework proposed by Seal et al. [50] is reduced by 79.7% compared to cloud platforms. Compared with the greedy power conservation algorithm [96] (GPC) and the greedy priority departure algorithm (GPDV), the DQN-based scheduling algorithm proposed by Atallah et al. [65] reduces the average delay by 10.2% and 21.1%, respectively.

Lan et al. [91] compared the average processing latency of all schemes. In the DDPG and offload-only schemes, as the computing power of the fog server increased, the delay caused by task processing decreased. Our proposed scheme reduces latency by approximately 18.8% compared to the offload-only scheme. This is because the offload-only processing model does not take into account the impact of service caching on network performance.

### 5.3. QOE

QOE reflects the gap between the actual situation and user expectations. Compared with the unicast-based routing algorithm, the cache hit ratio (CHR) of the DBN-based IRA proposed by Zhang et al. [27] increases by 19%. Compared with the traditional AC algorithm [61] (TACA), the standard policy gradient-based resource allocation algorithm (SPGRA) and the non-learning scheme [97], the soft-ACDRL algorithm proposed by Fang et al. [80] converges faster, and the performance improvement related to video quality is close to 15.2%, 16.5% and 32.5%, respectively.

### 5.4. Accuracy, Recall, Precision and F-Measure

Accuracy, recall, precision and F-measure are indispensable parts of evaluations of target detection models, vehicle division models and so on. First, we defined the binary prediction categories as follows:

1.	True Positive (TP): the positive result is predicted to be positive;
2.	True Negative (TN): the negative result is predicted to be negative;
3.	False Positive (FP): the negative result is predicted to be positive;
4.	False Negative (FN): the negative result is predicted to be negative.

The corresponding operation formula is given as follows:

$$ACC = \frac{TP + TN}{total} \tag{1}$$

$$REC = \frac{TP}{TP + FN} \tag{2}$$

$$PEC = \frac{TP}{TP + FP} \tag{3}$$

$$F = \frac{(\alpha^2 + 1) PEC \cdot REC}{\alpha^2 (PEC + REC)} \tag{4}$$

where $\alpha$ is the adjustable parameter.

Accuracy is the most common performance evaluation metric; it can describe the quality of the prediction model. Recall and precision usually need to be considered comprehensively, so a comprehensive evaluation metric F-measure is proposed which is the weighted sum average of the recall and precision, and its value is proportional to the performance of the model. In this paper, the performance and characteristics of the representative methods based on deep learning over CVIS are summarized in Table 2.

**Table 2.** Performance evaluation and characteristics of representative methods based on deep learning. Let '-' indicate that the data was not available.

| Author | Accuracy | Recall | Precision | F-Measure | Method |
|--------|----------|--------|-----------|-----------|--------|
| Chen [36] | 0.8 | - | - | - | DL. |
| Kumar [37] | 0.98 | - | - | 0.87 | R-CNN |
| Priyadharshini [47] | 0.964 | - | - | - | R-CNN |
| Huang [55] | 0.8 | - | - | - | YOLO |
| Jeon [38] | 0.9 | - | - | - | CNN and LSTM |
| Akilan [39] | - | - | - | 0.95 | MvRF-CNN |
| Ma [41] | - | - | 0.7364 | - | CNN |
| Jeong [43] | - | 0.9949 | - | 0.9704 | CNN |
| Seal [50] | - | - | 0.515 | - | YOLOv3 |
| Kamran [48] | - | - | 0.6279 | - | R-CNN |
| Sreekumar [54] | - | - | 0.71 | - | YOLOv2 |
| Humberto [52] | - | - | 0.579 | - | YOLOv3 |

## 6. Discussion

Up to now, there is little research on solving traditional problems based on deep learning, and the field of video application over CVIS is still in its infancy. Compared with the traditional methods, the video application method based on deep learning has lower average delay, higher QOE and more accurate detection and is more robust to the complex CVIS environment. Therefore, the use of deep learning algorithms to solve video business problems over CVIS is worthy of extensive research. The development trend of this research in the future is discussed as follows:

1. In order to restore the real environment and reflect more real data characteristics when building a model based on deep learning, it is a problem worthy of in-depth study to integrate the traffic characteristics such as the high-speed mobility of vehicles and the short life of channel links;

2. Due to the access environment of workshop communication being open, it will face the risk of malicious attacks in the CVIS environment [98,99]. Therefore, the security of vehicle communication needs to be paid enough attention, and it is necessary to design algorithms based on deep learning to solve network security problems and avoid video content transmission errors, identity authentication failures and network intrusion;

3. When facing large-scale traffic datasets, the model based on deep learning is slightly insufficient in terms of training speed and precision, so how to improve the training speed on the premise of ensuring precision is a problem that needs to be paid attention, so as to improve the training efficiency of the deep learning model;

4. Because of the single training environment, the deep learning algorithm has poor model transfer in the complex and changeable CVIS environment. Therefore, improving the generalization ability of the deep learning model is also the direction of optimizing the deep learning algorithm.

To sum up, firstly, we summarized the current video application methods based on deep learning over CVIS, and then described and analyzed the shortcomings and related problems in the existing research; finally, we discussed the possible development trends of this research in the future. From the simulation results of most studies at present, the deep learning algorithm performs very well compared with the traditional algorithm, which still has the potential for optimization. Therefore, it can be predicted that solving traditional problems through deep learning algorithms will become the mainstream trend in video applications over CVIS.

**Author Contributions:** Conceptualization, L.D.; methodology, B.S.; validation, L.D, Y.J. and B.S. All authors have read and agreed to the published version of the manuscript.

**Funding:** This work is supported by the National Key Research and Development Program of China (2021YFB2601401), and the Natural Science Foundation of Shaanxi Province, China (2019JQ-264).

**Institutional Review Board Statement:** Not applicable.

**Informed Consent Statement:** Not applicable.

**Data Availability Statement:** The datasets used were obtained from public open-source datasets from: 1. NGSIM: https://data.transportation.gov/Automobiles/Next-Generation-Simulation-NGSIM-Vehicle-Trajector/8ect-6jqj (accessed on 26 April 2022); 2. COCO dataset: http://cocodataset.org; 3. automotive Ethernet intrusion dataset: https://doi.org/10.21227/1yr3-q009 (accessed on 26 April 2022); 4. MIO-TCD dataset: https://www.kaggle.com/yash88600/miotcd-dataset-50000-imagesclassification (accessed on 26 April 2022); 5. VOC2007: http://host.robots.ox.ac.uk/pascal/VOC/voc2007/index.html (accessed on 26 April 2022); and 6. VOC2012: http://host.robots.ox.ac.uk/pascal/VOC/voc2012/index.html (accessed on 26 April 2022).

**Conflicts of Interest:** The authors declare no conflict of interest.

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
