# Peer review of "Deep Learning for Video Application in Cooperative Vehicle-Infrastructure System: A Comprehensive Survey"

_applsci, doi:10.3390/app12126283_

Round 1

Reviewer 1 Report

This paper review some video application for cooperative vehicle infrastructure system mainly from the perspective of deep learning. Overall, the paper was well-written and it lacks of related survey in this filed. Some problem include

(1) It would be better to clearly explain the organization of the paper. The motivation and the challenges should be highligted.

(2) It can prepare a graph to describe the contect of the paper.

(3) Part of contents are not closely related to CVIS application.

Reviewer 2 Report

The authors present a cooperative vehicle infrastructure system survey of Deep Learning uses. This is a hot topic in world transportation and the development of intelligent traffic system in the future. 

Overall the paper is interesting and well presented, but I would like to point out some issues that I will list below:

- The authors should review the template.

- The authors should adjust the image quality and increase their size. Maybe an architecture or a flowchart would improve the work.

- The references do not have the correct format. They should be revised and must include de DOI. 

Reviewer 3 Report

The authors have submitted a manuscript a review paper about the application of deep learning in cooperative vehicle infrastructure system.

My comments:

1.) The quality of figures is low. I think a scientific paper has to contain figures with much higher quality.

2.) In Table 2, there are many empty fields. If the data is not available, the authors should denote it.

3.) On which databases were the examined method evaluated? It would be good to read something about the publicly available benchmark databases with links from which they can be downloaded.

4.) Sample results, detections would be nice. This could illustrate the manuscript well.

5.) It would be interesting to read something about how image quality influences the performance of deep algorithms. In the literature, it is a hot research topic in the context of vehicles, since severe weather conditions can occur. Papers for citations: https://www.researchgate.net/publication/224258668_Applications_of_Objective_Image_Quality_Assessment_Methods/link/5a08bedeaca272ed279ff51d/download , https://www.mdpi.com/2076-3417/12/1/101 , https://arxiv.org/pdf/1608.05246.pdf?ref=https://githubhelp.com .

Round 2

Reviewer 3 Report

I think this manuscript can be accepted. This study reviews the state-of-the-art carefully and provides many useful background information for researchers active in this field.